# CONCORD: CONCEPT-INFORMED DIFFUSION FOR DATASET DISTILLATION

## ABSTRACT

Dataset distillation has witnessed significant progress in synthesizing small-scale datasets that encapsulate rich information from large-scale original ones. Particularly, methods based on generative priors show promising performance, while maintaining computational efficiency and cross-architecture generalization. However, the generation process lacks explicit controllability for each sample. Previous distillation methods primarily match the real distribution from the perspective of the entire dataset, whereas overlooking conceptual completeness at the instance level. This oversight can result in missing or incorrectly represented object details and compromised dataset quality. To this end, we propose to incorporate the conceptual understanding of large language models (LLMs) to perform a CONCept-infORmed Diffusion process for dataset distillation, in short as CONCORD. Specifically, distinguishable and fine-grained concepts are retrieved based on category labels to explicitly inform the denoising process and refine essential object details. By integrating these concepts, the proposed method significantly enhances both the controllability and interpretability of the distilled image generation, without replying on pre-trained classifiers. We demonstrate the efficacy of CONCORD by achieving state-of-the-art performance on ImageNet-1K and its subsets. It further advances the practical application of dataset distillation methods. The code implementation is attached in the supplementary material.

## 1 INTRODUCTION

In the current digital era, vast volumes of data are produced and disseminated across online platforms on a daily basis. The abundance of data boosts the training of robust neural network models, which often outperform human experts in a variety of domains (He et al., 2016; Dosovitskiy et al., 2022; Brown et al., 2020; Deng et al., 2009; Devlin et al., 2018). However, the heavy dependence on data also causes unbearable burden on the storage space and computational consumption. Strong neural networks often demand days or even months of training on high-capacity hardware, and this issue is exacerbated for more complex foundation models (Radford et al., 2021; He et al., 2022; Touvron et al., 2023; Bai et al., 2023). While pre-trained models are mostly available for general use, developing new networks from scratch remains necessary for certain specialized domains, and would be particularly challenging for resource-constrained research teams. In this context, Dataset Distillation (DD) emerges as a solution to condense rich information from original large-scale datasets into much smaller surrogate datasets (Wang et al., 2018; Zhao et al., 2021; Yu et al., 2023; Sachdeva & McAuley, 2023). With substantially reduced training time, the surrogate datasets aim to restore the performance levels of the original data for practical applications.

Typical DD methods incorporate meta-learning or metric matching to condense rich information into surrogate sets, and have achieved considerable performance on various benchmarks (Wang et al., 2018; Zhao et al., 2021; Nguyen et al., 2021b; Loo et al., 2022; Kim et al., 2022b). However, the distillation phase itself often demands even longer time compared with the training process on the original dataset (Cui et al., 2023; Sun et al., 2024). It would still be impractical for individual researchers to perform distillation on personalized datasets. Besides, these methods are easily biased towards the architecture adopted in the distillation phase, necessitating specialized designs to mitigate cross-architecture generalization challenges (Zhou et al., 2023; Wang et al., 2023a).

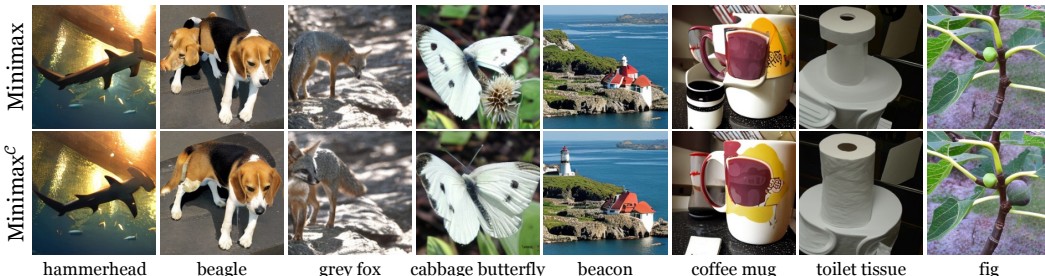

Figure 1: Comparison on example generated images with and without our proposed CONCORD method. $^{\mathcal{C}}$ indicates that CONCORD is applied. Incorporating rich knowledge from LLMs, CONCORD refines instance-level conceptual completeness, and enhances the overall dataset quality.

Recently, a series of methods integrate generative models to synthesize training data (Cazenavette et al., 2023; Gu et al., 2024a; Su et al., 2024; Moser et al., 2024). The pre-acquired generative prior within these models contributes to better cross-architecture generalization as well as significantly lower distillation consumption. While the synthetic images yield state-of-the-art performance, the distillation process lacks explicit controllability for each sample. Most existing approaches condense information by mimicking the distribution of real data at the dataset level. On the one hand, the lack of instance-level control might result in conceptual incompleteness, where essential object details may be missing or inaccurately represented in the generated images. Due to the constrained storage budget typical of DD benchmarks, this information loss cannot be sufficiently compensated. On the other hand, the distribution imitation is difficult to interpret, as the dataset quality can only be measured indirectly through training performance. It also raises a question: *does merely imitating the real distribution suffice for generating effective surrogate datasets?*

To this end, we intend to explicitly enhance instance-level conceptual completeness during the diffusion process with the assistance of large language models (LLMs). LLMs have obtained extensive conceptual understanding across a variety of objects, which can be utilized to facilitate examining and refining the defects and incorrect concept representations in the images. Our method involves initially retrieving distinguishable concepts specific to the target categories, and subsequently performing the CONCept-infORmed Diffusion inference (in short as CONCORD) to supplement missing or incorrect details. The approach offers several advantages. Firstly, the fine-grained control exerted by the retrieved concepts allows for more accurate refinement of object details, which also enables higher levels of personalization. Secondly, the concepts provide explicit explanations why the generated images are better suited for model training. Additionally, we employ concepts from similar categories to construct negative samples, thereby ensuring more accurate and stabilized control over the generation process. By prioritizing the enhancement of crucial concepts in addition to distribution imitating, our method generates more effective distilled data for training models.

As shown in Fig. 1, the samples generated by Minimax (Gu et al., 2024a) often fail to include complete and correct concepts for their respective categories, *e.g.*, unrealistic back legs of the beagle and a missing wing in the cabbage butterfly image. With the assistance of rich knowledge from LLMs, the proposed CONCORD method significantly improves the conceptual completeness, and reduces image defects. The proposed CONCORD method can be plugged into any diffusion-based generative pipelines for dataset distillation. We conduct extensive experiments on both Minimax and Stable Diffusion baselines (Ramesh et al., 2022) to illustrate the superiority of CONCORD, which achieves state-of-the-art performance on the full ImageNet-1K dataset and its subsets. Notably, the method only incorporates descriptive concepts to inform the diffusion process, eliminating the dependence on pre-trained classifiers. It reduces the required computational consumption, and thereby enhances the practicality of our approach for broader application possibilities.

## 2 RELATED WORK

**Dataset Distillation**  Aiming at reducing the demanded storage and computational consumption for training neural networks, dataset distillation (DD) has been increasingly investigated in recent

years (Yu et al., 2023; Sachdeva & McAuley, 2023) and achieved broad applications (Gu et al., 2024b; Xiong et al., 2023; Maekawa et al., 2024; Wang et al., 2023b). DD synthesizes small-scale datasets reflecting rich information from the original large-scale ones and is firstly designed with meta-learning schemes (Wang et al., 2018; Nguyen et al., 2021b;a; Zhou et al., 2022; Loo et al., 2022; 2023). The optimization is conducted upon a meta loss where a neural network or estimation is built on the surrogate data and then evaluated on the real data. Other methods optimize the synthetic images by matching training characteristics with real images (Zhao et al., 2021; Zhao & Bilen, 2023; Liu et al., 2023; Vahidian et al., 2024; Cazenavette et al., 2022; Zhao et al., 2023). The imitation on real distribution effectively improves the information contained in small surrogate datasets. Data parametrization (Kim et al., 2022b; Liu et al., 2022; Wei et al., 2024) and generative prior (Cazenavette et al., 2023; Gu et al., 2024a; Wang et al., 2023a) are also considered for more efficient DD method construction. However, most of existing DD methods remain as black boxes, lacking the ability of explicitly controlling the distilling direction. As a result, the practicality of DD methods are still poor from real-world applications. In this work, we aim at enhancing both the interpretability and controllability of the dataset distillation process.

**Diffusion Models**  Diffusion models have acquired substantial success in generating high-quality images (Ho et al., 2020; Dhariwal & Nichol, 2021; Kingma et al., 2021; Nichol & Dhariwal, 2021). There have also been a series of works focusing on diffusion-based image manipulating or editing. DiffusionCLIP incorporates a CLIP model into the diffusion model fine-tuning to provide optimization guidance (Kim et al., 2022a). DiffuseIT, DiffEdit and Prompt-to-Prompt integrate the editing into manifold constraint, mask guidance and cross attention control, respectively (Kwon & Ye, 2023; Couairon et al., 2023; Hertz et al., 2023). However, most of them manipulate image instances following certain instructions. SDEdit proposes to control the training data generation, yet it requires the assistance of pre-trained models (Yeo et al., 2024). In this work, we design a training-free denoising guidance towards images suitable for model training.

## 3  METHOD

In this section, we demonstrate the detailed modules of our proposed CONCept-infORmed Diffusion method (CONCORD). Firstly, we present the preliminary knowledge on dataset distillation and the possibility of performing concept-informed diffusion in Sec. 3.1. Subsequently we illustrate the design of concept acquirement and objective design in Sec. 3.2 and Sec. 3.3, respectively.

### 3.1  CONCEPT-INFORMED DIFFUSION

Given a target real dataset $\mathcal{T} = \{(\mathbf{x}_i, y_i)\}_{i=1}^{|\mathcal{T}|}$, the aim of dataset distillation is to generate a small surrogate dataset $\mathcal{S} = \{(\hat{\mathbf{x}}_i, y_i)\}_{i=1}^{|\mathcal{S}|}$, where $|\mathcal{S}| \ll |\mathcal{T}|$, such that training a network on $\mathcal{S}$ approximates as closely as possible the performance attained when training on $\mathcal{T}$. Typical methods incorporate meta-learning or metric matching to condense the information from real data into the surrogate dataset. However, the dependence on bi-level optimization often leads to excessive computation demands and bias towards specific adopted architectures (Sun et al., 2024; Zhou et al., 2023). Recently, methods utilizing the generative priors of diffusion models emerge as solutions for more efficient dataset distillation (Gu et al., 2024a; Su et al., 2024; Moser et al., 2024).

**Diffusion for Distillation**  Diffusion-based generative models learn data distributions via denoising. Firstly, a forward process is defined by obtaining $\mathbf{x}^{(T)}$ from clean data $\mathbf{x}^{(0)} \sim q(\mathbf{x}^{(0)})$ as a Markov chain of gradually adding Gaussian noise at time steps $t$ (Ho et al., 2020):

$$q(\mathbf{x}^{(1:T)}|\mathbf{x}^{(0)}) := \prod_{t=1}^{T} q(\mathbf{x}^{(t)}|\mathbf{x}^{(t-1)}), \text{ where } q(\mathbf{x}^{(t)}|\mathbf{x}^{(t-1)}) := \mathcal{N}(\mathbf{x}^{(t)}; \sqrt{1-\beta_t}\mathbf{x}^{(t-1)}, \beta_t\mathbf{I}), \quad (1)$$

where $\beta_t \in (0, 1)$ is a variance schedule. Denoting $\alpha_t := 1 - \beta_t$ and $\bar{\alpha}_t := \prod_{s=0}^{t} \alpha_s$, $\mathbf{x}^{(t)}$ at an arbitrary time step $t$ can be directly sampled with a Gaussian noise $\epsilon \sim \mathcal{N}(0, \mathbf{I})$:

$$\mathbf{x}^{(t)} = \sqrt{\bar{\alpha}_t}\mathbf{x}^{(0)} + \sqrt{1-\bar{\alpha}_t}\epsilon. \quad (2)$$

Denoising diffusion probablistic models (DDPMs) approximate the data distribution with a network:

$$p_\theta(\mathbf{x}^{(t-1)}|\mathbf{x}^{(t)}) = \mathcal{N}(\mathbf{x}^{(t-1)}; \mu_\theta(\mathbf{x}^{(t)}, t), \Sigma_\theta(\mathbf{x}^{(t)}, t)), \quad (3)$$

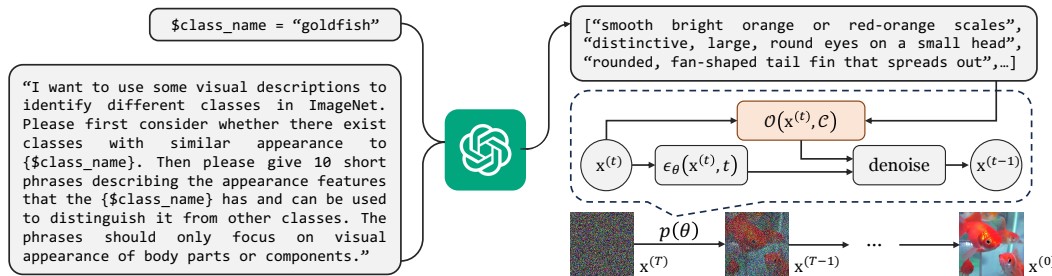

Figure 2: The pipeline of the proposed CONCORD method. Descriptive concepts are retrieved and utilized to inform the diffusion denoising process. The samples with better instance-level concept completeness help to construct a surrogate dataset with better overall quality.

where

$$\mu_\theta(\mathbf{x}^{(t)}, t) := \frac{1}{\sqrt{\alpha_t}} \left( \mathbf{x}^{(t)} - \frac{1 - \alpha_t}{\sqrt{1 - \bar{\alpha}_t}} \epsilon_\theta(\mathbf{x}^{(t)}, t) \right), \tag{4}$$

and the $\epsilon_\theta(\mathbf{x}^{(t)}, t)$ is the predicted noise, where $\theta$ is optimized by:

$$\min_\theta \mathbb{E}_{t, \mathbf{x}^{(0)} \sim q(\mathbf{x}^{(0)}), \epsilon \sim \mathcal{N}(0, \mathbf{I})} \left[ \| \epsilon - \epsilon_\theta(\sqrt{\bar{\alpha}_t} \mathbf{x}^{(0)} + \sqrt{1 - \bar{\alpha}_t} \epsilon, t) \|^2 \right]. \tag{5}$$

By denoising a fixed number of random noises, a surrogate dataset can be generated that encapsulates the distribution of original data. Gu et al. (2024a) introduce additional minimax criteria to distill more representative and diverse samples from real data, improving the quality of the generated surrogate datasets. However, the distilling process primarily focuses on imitating dataset-level concept distributions, while overlooking the instance-level conceptual completeness at the inference stage. Since the final distilled samples are directly derived from random noise, without explicit control over the content, essential object details might be missing or incorrectly represented in the generated images. Moreover, the constrained storage budget typical of dataset distillation benchmarks limits the ability to compensate the instance-level information loss by increasing data scale, further compromising the quality of the distilled dataset. Thus, there is an urgent demand for techniques that allow for explicit control during the denoising process, enhancing both the conceptual completeness and the overall quality of the surrogate dataset.

**Concept Informing** Dhariwal & Nichol (2021) introduce classifier guidance with the gradients of a classifier network $\nabla_{\mathbf{x}^{(t)}} \log p_\phi(y|\mathbf{x}^{(t)}, t)$ during the diffusion process. However, when the classifier can acquire activations from a broad set of possible details to make predictions, the conceptual completeness associated with the specific category can remain insufficient. Therefore, we propose to explicitly inform the diffusion process with fine-grained and distinguishable concepts tied to the category (*e.g.*, attributes). The concept-informed diffusion offers several advantages: firstly, various concepts of a category provide more detailed information compared with using the category alone, allowing for explicit reasoning and refinement during the generation process. Secondly, on circumstances where classifiers are difficult to obtain, concepts remain viable given the category. We define the set of concepts associated with the category label of the current sample $\mathbf{x}^{(t)}$ as $\mathcal{C} = \{a_j\}_{j=1}^{|\mathcal{C}|}$, where $|\mathcal{C}|$ is a pre-defined number of concepts. Subsequently, an objective $\mathcal{O}(\mathbf{x}^{(t)}, \mathcal{C})$ can be derived reflecting the semantic similarity between the generated sample and these concepts. The informed denoising process can be represented with the objective as:

$$p_\theta(\mathbf{x}^{(t-1)}|\mathbf{x}^{(t)}) = \mathcal{N}(\mathbf{x}^{(t-1)}; \mu_\theta(\mathbf{x}^{(t)}, t) + \Sigma_\theta(\mathbf{x}^{(t)}, t) \nabla_{\mathbf{x}^{(t)}} \mathcal{O}(\mathbf{x}^{(t)}, \mathcal{C}), \Sigma_\theta(\mathbf{x}^{(t)}, t)), \tag{6}$$

Song et al. (2020) introduce another form of denoising diffusion implicit models (DDIMs) that construct a deterministic non-Markovian inference process as:

$$\mathbf{x}^{(t-1)} = \sqrt{\bar{\alpha}_{t-1}} \hat{\mathbf{x}}^{(0)} + \sqrt{1 - \bar{\alpha}_{t-1}} \cdot \epsilon_\theta(\mathbf{x}^{(t)}, t), \tag{7}$$

where the estimated observation $\hat{\mathbf{x}}^{(0)}$ of clean original data $\mathbf{x}^{(0)}$ can be obtained by computing the posterior expectation with $\mathbf{x}^{(t)}$ (Robbins, 1992):

$$\hat{\mathbf{x}}^{(0)} := \frac{\mathbf{x}^{(t)}}{\sqrt{\bar{\alpha}_t}} - \frac{\sqrt{1 - \bar{\alpha}_t}}{\sqrt{\bar{\alpha}_t}} \epsilon_\theta(\mathbf{x}^{(t)}, t). \tag{8}$$

---

**Algorithm 1:** Concept-Informed Diffusion

---

**Input:** diffusion model $\theta$, original dataset $\mathcal{T}$, concept set $\{\mathcal{C}\}$, required sample number $N_{\mathcal{S}}$
**Output:** surrogate dataset $\mathcal{S}$
Initialize the surrogate dataset $\mathcal{S} = \{\}$
**for** *index in 1...$N_{\mathcal{S}}$* **do**
    Obtain a random noisy sample $\mathbf{x}^{(T)}$ and a category label $c$
    Retrieve the positive and negative concepts $\mathcal{C}$ and $\tilde{\mathcal{C}}$ from $\{\mathcal{C}\}$ according to label $c$
    **for** *time step $t$ in T...1* **do**
        Predict the noise $\epsilon_\theta(\mathbf{x}^{(t)}, t)$
        Calculate the concept matching objective $\mathcal{O}(\mathbf{x}^{(t)}, \mathcal{C}, \tilde{\mathcal{C}})$ according to Eq. 12
        Update the predicted noise $\hat{\epsilon}$ according to Eq. 9
        Conduct denoising step to obtain $\mathbf{x}^{(t-1)}$ according to Eq. 7
    **end**
    Add the predicted clean sample to the surrogate set $\mathcal{S} \leftarrow \mathbf{x}^{(0)}$
**end**

---

Similarly, we can apply the concept informing through:

$$\hat{\epsilon} := \epsilon_\theta(\mathbf{x}^{(t)}, t) - \lambda\sqrt{1 - \bar{\alpha}_t} \nabla_{\mathbf{x}^{(t)}} \mathcal{O}(\mathbf{x}^{(t)}, \mathcal{C}), \tag{9}$$

where $\lambda$ is the informing weight adjustable for control extent. The updated $\hat{\epsilon}$ is subsequently used for the above reverse diffusion process. When the informing can be applied to both frameworks, in this work, we mainly incorporate DDIM for developing our algorithm.

## 3.2 CONCEPT ACQUIREMENT

Based on the aim of enhancing the discriminative details and mitigating conceptual incompleteness in the generated images, we intend to explicitly inform the diffusion process with distinguishable concepts. While concluding or manually designing visual concepts being infeasible for a large number of categories, large language models (LLMs) offer a valuable solution with rich conceptual understanding acquired during the training process. Inspired by this, we design prompts for the corresponding categories in the target dataset to elicit fine-grained attributes from LLMs, which are used as concepts to inform the diffusion process. Menon & Vondrick (2023) design prompts to retrieve descriptions used for zero-shot image classification. While our task shares certain similarity, it differs primarily in the nature of the required descriptions. Descriptions used for classification should comprehensively reflect various aspects of the corresponding category. In comparison, those used for constructing surrogate datasets are supposed to be distinguishable across different categories to ensure that the generated data can facilitate model training. Therefore, we design an example prompt shown in Fig. 2, where distinction from other classes is emphasized for retrieving descriptions.

**Concept Validity Evaluation** Once a set of concepts is obtained, it is crucial to evaluate their validity on actual data, as some concepts may not be well-represented in the real data due to biases in data collection. Thus, before the concept matching process, we first retrieve an over-abundant amount of concepts, and then filter them through a validity evaluation process to identify those with the strongest relevance to the real data. For this purpose, we utilize a CLIP model to extract embedded features from both real images and textual descriptions. For a category $l$, the activation $A$ of a text description $c$ on the images $\{\mathbf{x}_i; y_i = l\}_{i=1}^{|l|}$ can be calculated by:

$$A = \left\langle \frac{1}{|l|} \sum_{i=1}^{l} \psi(\mathbf{x}_i), \psi(c) \right\rangle, \tag{10}$$

where $\psi(\cdot)$ denotes the embedded feature extraction function of the CLIP model, and $\langle \cdot, \cdot \rangle$ computes the cosine similarity. We select a pre-defined number of $|\mathcal{C}|$ descriptions for each category with the highest activation scores for further use in the informing process. This ensures that the selected concepts retain the integrity of the knowledge distilled from the real dataset, making them more representative and relevant for improving instance-level conceptual completeness in generated data.

### 3.3 CONCEPT MATCHING

With the distiguishable concepts acquired, a straightforward approach to measure the relationship between the generated sample $\mathbf{x}^{(t)}$ and the corresponding concepts $\mathcal{C} = \{c_j\}_{j=1}^{|\mathcal{C}|}$ is to compute their cosine similarity:

$$\mathcal{O}(\mathbf{x}^{(t)}, \mathcal{C}) = -\frac{1}{|\mathcal{C}|} \sum_{i=1}^{|\mathcal{C}|} \left\langle \psi(\mathbf{x}^{(t)}), \psi(c_i) \right\rangle, \tag{11}$$

which is similar to the concept validity evaluation process. We argue that beyond the positive informing from the concepts associated with the corresponding category, it is equally important to adjust the diffusion control by considering the overall dataset distribution. Therefore, we employ concepts from other categories as negative samples to provide more stable diffusion guidance.

**Contrastive Matching**   Inspired by the contrastive loss adopted in CLIP training (Radford et al., 2021; Patel et al., 2023), we propose a similar strategy to incorporate negative concepts. Since multiple positive concepts should work together to provide adequate guidance, we modify the supervised contrastive loss (Khosla et al., 2020) into an image-text version:

$$\mathcal{O}(\mathbf{x}^{(t)}, \mathcal{C}, \tilde{\mathcal{C}}) = -\frac{1}{|\mathcal{C}|} \sum_{i=1}^{|\mathcal{C}|} \log \left( \frac{\exp(\langle \psi(\mathbf{x}^{(t)}), \psi(c_i) \rangle / \tau)}{\exp(\langle \psi(\mathbf{x}^{(t)}), \psi(c_i) \rangle / \tau) + \sum_{a_j \in \tilde{\mathcal{C}}} \exp(\langle \psi(\mathbf{x}^{(t)}), \psi(c_j) \rangle / \tau)} \right), \tag{12}$$

where $\tilde{\mathcal{C}}$ denotes the set of negative concepts.

**Negative Concept Selection**   With a large number of potential negative categories, selecting appropriate negative concepts is essential for effectively informing the diffusion process. We first compute the cosine similarity between the category labels, and use the similarity as sampling weight for negative category selection. This approach ensures that categories with higher similarity to the target category are prioritized as negative samples. Compared with random selection, the similarity-based approach offers more precise control over the diffusion process. Additionally, compared with a fixed range of negative categories, the dynamic sampling allows for more diverse denoising control.

## 4 EXPERIMENTS

### 4.1 IMPLEMENTATION DETAILS

We adopt Minimax (Gu et al., 2024a) and Stable Diffusion unCLIP Img2Img (Ramesh et al., 2022) as baselines to evaluate our proposed training-free approach, which is applied at the inference stage. The informing weight $\lambda$ in Eq. 9 is set as 1. For each category, 5 descriptive attributes are selected with the highest activation scores, as detailed in Sec. 3.2. And 10 negative descriptions from different categories are used for contrastive loss calculation. A total denoising step number of 50 is adopted for the generation process, and the generated images are resized to $224 \times 224$ for subsequent validation. The validation protocol follows RDED (Sun et al., 2024), where soft label is adopted to obtain better performance. The model training lasts for 300 epochs. All reported results are based on 3 random runs, with the averaged accuracy and the variance included. All the experiments are conducted on a single NVIDIA A100 GPU. Further implementation details are provided in Sec. B.

We believe that DD for small-resolution datasets has been well solved by previous methods. Thus, the main experiments in this work are conducted on ImageNet-1K (Deng et al., 2009) and its sub-sets including ImageNet-100 and ImageWoof (Fastai). Additionally, we incorporate Food-101 (Bossard et al., 2014) as another benchmark to evaluate the effectiveness of the proposed CONCORD method.

### 4.2 COMPARISON WITH STATE-OF-THE-ARTS

Firstly we conduct the experiments on standard benchmarks, reporting the performance on multiple different architectures. The compared methods include MTT (Cazenavette et al., 2022), SRe$^2$L (Yin et al., 2023), RDED (Sun et al., 2024), DiT (Peebles & Xie, 2023), Minimax (Gu et al., 2024a), and Img2Img (Ramesh et al., 2022). The results on ImageWoof, ImageNet-100, and the full ImageNet-1K are shown in Tab. 1 and Tab. 2, respectively.

Table 1: Performance comparison with state-of-the-art methods on ImageWoof. The superscript $^{\mathcal{C}}$ indicates the application of our proposed CONCORD method. **Bold** entries indicate best results, and underlined ones illustrate improvement over baseline.

| IPC (Ratio) | Test Model | MTT | SRe$^2$L | RDED | DiT | Minimax | Minimax$^{\mathcal{C}}$ | unCLIP | unCLIP$^{\mathcal{C}}$ |
|---|---|---|---|---|---|---|---|---|---|
| 1 (0.08%) | ConvNet | **28.6**$_{\pm0.8}$ | - | 18.5$_{\pm0.9}$ | 20.5$_{\pm0.8}$ | 16.7$_{\pm0.2}$ | 17.8$_{\pm0.8}$ | 20.5$_{\pm0.4}$ | 19.9$_{\pm0.7}$ |
| | ResNet-18 | - | 13.3$_{\pm0.5}$ | **20.8**$_{\pm1.2}$ | 18.3$_{\pm0.7}$ | 15.3$_{\pm1.1}$ | 16.9$_{\pm1.0}$ | 16.7$_{\pm0.7}$ | 17.4$_{\pm1.1}$ |
| | ResNet-101 | - | 13.4$_{\pm0.1}$ | **19.6**$_{\pm1.8}$ | 17.1$_{\pm1.3}$ | 14.2$_{\pm1.1}$ | 14.9$_{\pm1.3}$ | 14.9$_{\pm0.2}$ | 15.3$_{\pm1.3}$ |
| 10 (0.8%) | ConvNet | 35.8$_{\pm1.8}$ | - | 40.6$_{\pm2.0}$ | 42.2$_{\pm1.2}$ | 41.2$_{\pm0.8}$ | **43.1**$_{\pm0.5}$ | 40.1$_{\pm0.8}$ | 41.2$_{\pm0.8}$ |
| | ResNet-18 | - | 20.2$_{\pm0.2}$ | 38.5$_{\pm2.1}$ | 38.2$_{\pm1.1}$ | 42.8$_{\pm1.1}$ | **44.4**$_{\pm0.9}$ | 37.9$_{\pm1.1}$ | 40.7$_{\pm0.4}$ |
| | ResNet-101 | - | 17.7$_{\pm0.9}$ | 31.3$_{\pm1.3}$ | 31.1$_{\pm0.3}$ | 35.7$_{\pm0.9}$ | **36.5**$_{\pm0.9}$ | 30.7$_{\pm0.9}$ | 31.9$_{\pm1.1}$ |
| 50 (3.8%) | ConvNet | - | - | 61.5$_{\pm0.3}$ | 59.9$_{\pm0.2}$ | 61.1$_{\pm0.8}$ | **62.5**$_{\pm0.9}$ | 59.5$_{\pm1.4}$ | 60.4$_{\pm0.4}$ |
| | ResNet-18 | - | 23.3$_{\pm0.3}$ | 68.5$_{\pm0.7}$ | 65.9$_{\pm0.2}$ | 67.8$_{\pm0.5}$ | **69.2**$_{\pm1.0}$ | 63.6$_{\pm0.6}$ | 66.1$_{\pm1.1}$ |
| | ResNet-101 | - | 21.2$_{\pm0.2}$ | 59.1$_{\pm0.7}$ | 60.1$_{\pm1.1}$ | 62.2$_{\pm0.6}$ | **63.6**$_{\pm0.2}$ | 60.0$_{\pm1.0}$ | 60.8$_{\pm0.9}$ |

Table 2: Performance comparison with state-of-the-art methods on ImageNet-100 (left) and ImageNet-1K (right). The superscript $^{\mathcal{C}}$ indicates the application of our proposed CONCORD method. **Bold** entries indicate best results, and underlined ones illustrate improvement over baseline.

| Method | IPC | | | Method | IPC | | |
|---|---|---|---|---|---|---|---|
| | 1 | 10 | 50 | | 1 | 10 | 50 |
| SRe$^2$L | 3.0$_{\pm0.3}$ | 9.5$_{\pm0.4}$ | 27.0$_{\pm0.4}$ | SRe$^2$L | 0.1$_{\pm0.1}$ | 21.3$_{\pm0.6}$ | 46.8$_{\pm0.2}$ |
| RDED | 8.1$_{\pm0.3}$ | **36.0**$_{\pm0.3}$ | 61.6$_{\pm0.1}$ | RDED | **6.6**$_{\pm0.2}$ | 42.0$_{\pm0.1}$ | 56.5$_{\pm0.1}$ |
| DiT | **8.2**$_{\pm0.1}$ | 29.5$_{\pm0.4}$ | 59.8$_{\pm0.5}$ | DiT | 6.1$_{\pm0.1}$ | 41.3$_{\pm0.3}$ | 56.6$_{\pm0.2}$ |
| Minimax | 5.8$_{\pm0.2}$ | 31.6$_{\pm0.1}$ | 64.0$_{\pm0.5}$ | Minimax | 6.0$_{\pm0.1}$ | 43.4$_{\pm0.3}$ | 59.1$_{\pm0.1}$ |
| Minimax$^{\mathcal{C}}$ | 7.1$_{\pm0.2}$ | 33.3$_{\pm0.6}$ | 64.9$_{\pm0.3}$ | Minimax$^{\mathcal{C}}$ | 6.4$_{\pm0.2}$ | **43.8**$_{\pm0.6}$ | **59.4**$_{\pm0.2}$ |
| unCLIP | 7.1$_{\pm0.1}$ | 26.9$_{\pm0.4}$ | 64.6$_{\pm0.2}$ | unCLIP | 5.9$_{\pm0.2}$ | 42.0$_{\pm0.3}$ | 58.1$_{\pm0.2}$ |
| unCLIP$^{\mathcal{C}}$ | 7.7$_{\pm0.2}$ | 28.1$_{\pm0.7}$ | **65.4**$_{\pm0.4}$ | unCLIP$^{\mathcal{C}}$ | 6.2$_{\pm0.3}$ | 42.5$_{\pm0.2}$ | 58.5$_{\pm0.1}$ |

Under the 1 Image-per-class (IPC) setting, previous methods MTT and RDED have demonstrated the best performance, with the vanilla DiT model also showing strong results. Minimax is fine-tuned to enhance the representativeness and diversity of the generated data. Although it is less effective under small IPC settings, the performance superiority is more substantial as the IPC increases. The unCLIP Img2Img model is not specifically trained or fine-tuned on ImageNet, but still yields comparable performance by direct inference. When the proposed CONCORD method is applied to both baseline methods, significant performance improvements are observed across all IPC settings and architectures. These results indicate that refining instance-level conceptual completeness is essential for constructing more effective distilled datasets. However, we can also notice that the performance gain is less significant as the class number increases. A potential explanation is that the influence of instance-level quality diminishes as the overall data scale is larger. Despite this, the proposed CONCORD method achieves state-of-the-art performance on the full ImageNet-1K dataset and its subsets, especially on large IPC settings, further supporting its effectiveness in dataset distillation.

Additionally, we conduct experiments on Food-101 with unCLIP Img2Img as the baseline in Tab. 3. It simulates actual DD application scenarios for custom datasets. The results suggest that methods based on generative prior are capable and practical to perform custom DD tasks without extra training efforts. While the unCLIP baseline performs worse than random selection under the 50-IPC setting, the proposed CONCORD method still enhances the quality of distilled datasets across all IPC settings. It opens up new possibilities for resource-limited researchers to perform custom DD.

## 4.3 ABLATION STUDY AND DISCUSSION

In this section we conduct component analysis and experimental results on extended settings. By default, the experiments are conducted on ImageWoof, with unCLIP Img2Img as the baseline.

**Prompt Design** We employ LLMs to retrieve essential visual descriptions as the informing target. The description quality is crucial for achieving optimal informing effects. Therefore, a quantita-

Table 3: Performance comparison with un-CLIP Img2Img on Food-101 dataset.

| Method | IPC | | |
|---|---|---|---|
| | 1 | 10 | 50 |
| Random | $5.0_{\pm 0.1}$ | $30.1_{\pm 0.1}$ | $\mathbf{64.0}_{\pm 0.2}$ |
| unCLIP | $6.4_{\pm 0.1}$ | $30.7_{\pm 0.2}$ | $61.3_{\pm 0.3}$ |
| unCLIP$^C$ | $\mathbf{6.9}_{\pm 0.1}$ | $\mathbf{32.0}_{\pm 0.2}$ | $\underline{62.5}_{\pm 0.2}$ |

Table 4: Comparison with different prompts for concept retrieval on ImageWoof.

| Method | IPC | | |
|---|---|---|---|
| | 1 | 10 | 50 |
| Classification | $\mathbf{17.6}_{\pm 2.0}$ | $38.2_{\pm 1.3}$ | $64.1_{\pm 0.7}$ |
| Ours-3.5 | $16.8_{\pm 0.5}$ | $38.8_{\pm 0.2}$ | $65.4_{\pm 1.1}$ |
| Ours-4 | $17.4_{\pm 1.1}$ | $\mathbf{40.7}_{\pm 0.4}$ | $\mathbf{66.1}_{\pm 1.1}$ |

Table 5: Comparison with different negative description selection on ImageWoof.

| Method | IPC | | |
|---|---|---|---|
| | 1 | 10 | 50 |
| Random | $15.5_{\pm 1.6}$ | $39.5_{\pm 1.2}$ | $64.9_{\pm 0.2}$ |
| Similar-10 | $16.1_{\pm 0.6}$ | $38.3_{\pm 0.9}$ | $65.3_{\pm 1.2}$ |
| Similar-25 | $15.9_{\pm 1.0}$ | $37.9_{\pm 0.4}$ | $64.5_{\pm 0.5}$ |
| Similar-50 | $15.9_{\pm 1.4}$ | $38.3_{\pm 1.2}$ | $64.6_{\pm 0.4}$ |
| Weighted | $\mathbf{17.4}_{\pm 1.1}$ | $\mathbf{40.7}_{\pm 0.4}$ | $\mathbf{66.1}_{\pm 1.1}$ |

Table 6: Ablation study on the optimization baseline and objectives on ImageWoof.

| Base | Objective | IPC | | |
|---|---|---|---|---|
| | | 1 | 10 | 50 |
| DiT | None | $18.3_{\pm 0.7}$ | $38.2_{\pm 1.1}$ | $65.9_{\pm 0.2}$ |
| | Contrastive | $\mathbf{20.3}_{\pm 0.7}$ | $\mathbf{40.5}_{\pm 1.2}$ | $\mathbf{67.6}_{\pm 0.4}$ |
| unCLIP | None | $16.7_{\pm 0.7}$ | $37.9_{\pm 1.1}$ | $63.6_{\pm 0.6}$ |
| | Classifier | $16.9_{\pm 0.7}$ | $38.5_{\pm 1.0}$ | $65.2_{\pm 0.8}$ |
| | Cosine | $\mathbf{18.2}_{\pm 1.6}$ | $39.7_{\pm 1.1}$ | $63.9_{\pm 0.2}$ |
| | Contrastive | $17.4_{\pm 1.1}$ | $\mathbf{40.7}_{\pm 0.4}$ | $\mathbf{66.1}_{\pm 1.1}$ |

tive comparison between different prompt design and LLM models is provided in Tab. 4. Menon & Vondrick (2023) design prompts to retrieve descriptions for zero-shot classification, denoted as "Classifition" in the table. While the retrieved concepts improve performance when IPC=1, the impact is less significant for larger IPCs. Accordingly, we design a new prompt (shown in Fig. 2) that emphasizes distinguishable appearance features. The descriptions are retrieved from GPT-3.5 and GPT-4, and the GPT-4 version achieves overall the best performance improvement. Detailed examples of the retrieved descriptions are shown in Fig. 7 for further investigation.

**Negative Description Selection** In the contrastive objective of Eq. 12, negative concepts are introduced for more accurate informing. While the extra constraint potentially brings more information, the selection of negative concepts is critical for stable optimization. Therefore, we evaluate the influence of different selection strategies in Tab. 5. Firstly, random selection from all categories considerably enhances the quality of the distilled dataset. Given that concepts from similar categories can serve as more challenging negative guidance, we narrow the random selection range to include only the top-similar categories, denoted as "Similar-#" with the number indicating the range. However, the unstable performance improvement suggests that limiting the diversity of negative concepts can harm the informing effect. Eventually, we propose to adopt a weighted sampling strategy based on category similarity. By simultaneously emphasizing similar categories and maintaining diversity, the strategy achieves the most significant and stable performance improvement.

**Optimization** We adopt a contrastive design of the objective to incorporate negative descriptions and stabilize the informing process. Accordingly, different objective forms are compared in Tab. 6 to evaluate the effectiveness of this design. After tuning, the informing weight for classifier guidance is set as 0.05 for best performance, which provides consistent improvement over the baseline. However, the supervision from a class-level is too coarse to refine the necessary details for sample generation. This limitation is evident in the superior performance achieved by the contrastive objective, which offers more detailed guidance. Additionally, the reliance on extra pre-trained classifiers also reduces the practicality of classifier guidance. Comparatively, the cosine objective in Eq. 11 yields even larger improvement when only 1 image is used for training. As the IPC grows, the performance improvement decreases, potentially due to limited diversity from only positive concepts. Since the proposed CONCORD method is designed to work without the need for pre-trained classifiers, we focus exclusively on concept informing in the main experiments.

We also conduct experiments on the vanilla DiT model without Minimax fine-tuning, where the top-1 accuracy improves by 2% across different IPCs. It further validates our hypothesis that instance-level conceptual completeness is essential for dataset distillation methods based on generative prior.

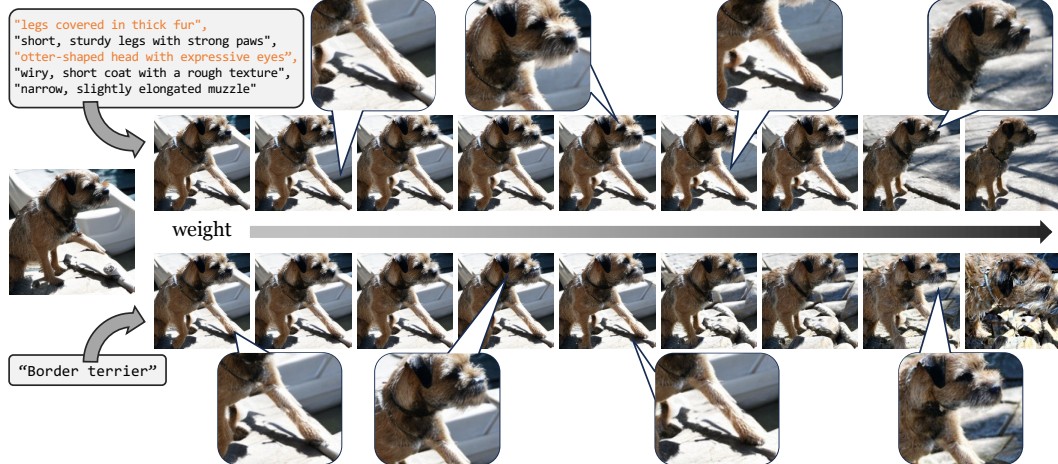

Figure 3: Comparison between images informed by fine-grained descriptive concepts (the first row) and the class name alone (the second row). From left to right the informing weight is gradually increased. Descriptions and corresponding image details are highlighted to illustrate better control with distinguishable concepts.

And the proposed CONCORD method can be broadly applied to existing diffusion pipelines to enhance the quality of the distilled datasets, which proves its practicality.

**Sample Visualization** We present example generated images with and without our proposed CONCORD method in Fig. 1. The baseline Minimax (Gu et al., 2024a) generates images with realistic texture and diverse variations. However, it overlooks the instance-level conceptual completeness, where essential concepts are often incorrect or missing (*e.g.*, the unnatural shape of the coffee mug and the absence of beacon in the beacon image). By applying the CONCORD method, the generated images demonstrate substantial improvement in representing essential object details. In dataset distillation, where the number of samples is limited, the instance-level defects can severely affect the quality of the distilled dataset. In contrast, by emphasizing conceptual completeness at the instance level, our proposed CONCORD method enhances the overall quality of generated samples, also providing interpretability for the superior performance.

**Effectiveness of Descriptions** We employ descriptive attributes generated by language models as concepts to inform the denoising process. In Fig. 3 we compare the informing effects using detailed descriptions versus class names. For avoiding the influence of objective forms, we perform the experiments using cosine similarity as in Eq. 11, and only match positive concepts. Several conclusions can be drawn from the comparison of results. Firstly, while class names provide certain level of concept understanding, fine-grained descriptions offer more precise control over the diffusion process. For instance, when informed by a description like "legs covered in thick fur", the length of the leg fur visibly increases as the informing weight grows, whereas images constrained by only the class name do not show a similar trend. Secondly, as the informing weight increases, images constrained by class names tend to collapse more quickly. It indicates that fine-grained concept informing provides better stability during the diffusion process compared with relying solely on class names. Thirdly, crucial descriptions such as "otter-shaped head with expressive eyes" effectively constrain the diffusion process. Even as images start to collapse, the head shape remains similar to the original generation result. In contrast, without explicit constraints from fine-grained descriptions, images informed by the class name show concept shift in these discriminative details.

**IPC Scale-up** An advantage offered by distillation methods based on generative prior is the flexibility to create surrogate datasets of varying sizes. Beyond the standard small-size benchmarks, we further extend the dataset size to 200 IPC in Fig. 4a. Across all IPC settings, the proposed CONCORD method provides consistent improvement upon both Minimax and unCLIP baselines. Notably, with 200 images per class, Minimax with CONCORD achieves the top-1 accuracy attained with the entire original ImageWoof dataset, following the same validation protocol.

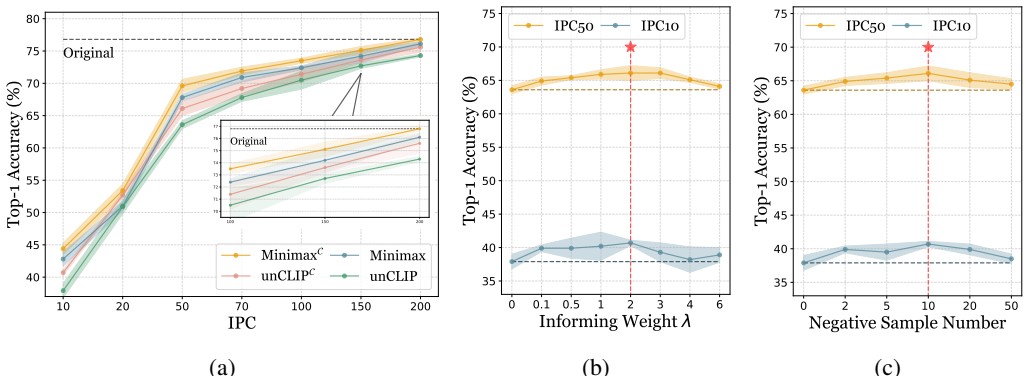

(a)          (b)          (c)

Figure 4: (a) Applying the proposed concept informing brings consistent improvement across all IPC settings. With 200 Images per class, our method achieves the performance attained with the full original set. (b/c) Parameter analysis on informing weight $\lambda$ and negative sample number.

## 4.4 PARAMETER ANALYSIS

There are multiple hyper-parameters involved in the proposed method. In this section, we perform analysis by adjusting the parameter to observe the influence on the performance.

**Informing Weight $\lambda$**   The informing weight controls the degree of influence applied to the denoising diffusion process. As shown in Fig. 4b, setting $\lambda = 0$ results in standard inference without concept informing. As $\lambda$ increases within a reasonable range, the performance is also improved, indicating that the injected concept information enhances the quality of distilled datasets. However, if $\lambda$ is too high, it disrupts the standard denoising process, leading to performance drop. Through comparison, we set the value of $\lambda$ as 2.0 for balance between sufficient control and stable denoising.

**Negative Sample Number**   The number of negative concepts is critical for constructing an effective contrastive loss. Therefore, we investigate the influence of negative sample number in Fig. 4c. When zero negative samples are used, cosine objective is applied for informing as in Eq. 11. Both too few or too many negative samples lead to unstable optimization and sub-optimal performance. Unlike standard contrastive learning, where the encoder separates different instances, the goal in DD is to focus on emphasizing essential object concepts. Therefore, enhancing positive concepts is more important. Based on our analysis, we adopt 10 negative samples in the contastive objective to provide an appropriate constraint while maintaining stable optimization.

## 5 CONCLUSION

In this work, we propose to incorporate the conceptual understanding of large language models (LLMs) to enhance instance-level image quality for dataset distillation. Specifically, distinguishable concepts are retrieved based on category labels, and are subsequently utilized to inform the diffusion-based sample generation process. The conceptual completeness obtained by the proposed CONCept-infORmed Diffusion (CONCORD) process mitigates the information loss caused by image defects, leading to higher overall quality of distilled datasets. CONCORD is evaluated on multiple baselines, and achieves state-of-the-art performance on the full ImageNet-1K dataset. The generated real-looking images with necessary details provide explicit interpretability for their effectiveness, and also prompt new possibilities of down-stream applications of dataset distillation.

**Limitations and Future Works**   The proposed concept informing method significantly improves the instance-level concept completeness, and thereby enhances the performance of the distilled data. But simultaneously, it also involves extra computational cost. Since the informing is conducted throughout the diffusion denoising process, the method might not be applicable to few-step diffusion techniques, which aim to reduce computational overhead. In future works, we will explore efficient diffusion inference techniques for more practical dataset distillation.

**Reproducibility Statement** We have provided implementation details regarding the baseline preparation, the proposed CONCORD method as well as the evaluation process in the Appendix Sec. B. We use the publicly available ImageNet dataset as well as its subsets for conducting experiments. Additionally, the utilized source code is attached in the supplementary material, and will be made public upon acceptance.

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

APPENDIX

The appendix is organized into the following sections: In Sec. A we provide additional justification from the literature as far as the utility of using LLM-based concept informed learning. In Sec. B we introduce the implementation details of our method. In Sec. C we present more experiment results and analysis on the proposed CONCORD method. In Sec. D we show example generated images to better illustrate the effect of the proposed concept informing method.

## A    FURTHER ANALYTICAL GROUNDING

**Perspective from XAI**    Explainable Artificial Intelligence (XAI) is an emerging area within machine learning that aims to provide users with greater insight into the functioning and mechanisms of black-box models, such as neural networks. Standard practice often involves knowledge extraction techniques, where broader, less precise, but simpler and thus more intuitive models are presented to explain the behavior of complex machine learning models. However, it has been noticed that this paradigm can be amended with the success and proliferation of LLMs (Ehsan et al., 2024). In particular, LLMs enable a more iterative and active learning procedure, where user prompts can can directly inform the learning process to accommodate user needs. Simultaneously, LLMs can periodically generate language-based explanations, offering updates on model progress and adjustments.

One of the key advantages of LLMs is their potential for personalization (Chen et al., 2024). Given the rich variety of concepts derived from the extensive training data and the depth of developed models, LLMs are capable to foster a detailed and more human-centric understanding. This allows the models to tune the learning process towards specific application-driven concerns. The effectiveness of LLMs as explainers (Kroeger et al., 2023) provides the clear potential to address important use case concerns that are difficult to represent through standard analytical loss functions. This adaptability allows LLMs to bridge gaps between the learning objectives and real-world applications.

**Perspective from Instrumental DD**    In the recent work by Kungurtsev et al. (2024), it has been argued that an important analytical consideration often overlooked in most optimization formulations for dataset distillation is its instrumentality. Specifically, synthetic data is typically not just used to solve the same learning problem in the same setting, but rather the dataset is expected to be used in some broader applications of interest to the user. These applications may have information needs that are not inherently condensed by standard off-the-shelf DD algorithms. By including additional custom criteria into the DD optimization formulation, while still incorporating existing powerful tools, DD can be more effectively steered towards performance on desired use cases. In this work, concepts are employed to facilitate natural human taxonomy with respect to object identification and recognition, and this consideration substantially improves the process by aligning the synthetic dataset with desired test performance outcomes.

## B    MORE IMPLEMENTATION DETAILS

**Baselines**    We adopt Minimax (Gu et al., 2024a) and Stable Diffusion unCLIP Img2Img (Ramesh et al., 2022) as the baselines to illustrate the efficacy of our proposed concept informing method. These two baselines represent two different application scenarios, as outlined below.

For Minimax, a fine-tuning process is conducted on ImageNet-1K. While fine-tuning on target datasets yields superior performance, it also demands more resource consumption. Additionally, class labels are utilized for conditioning the denoising process, which might be inconvenient when extending the model to broader datasets. We adopt the default parameter setting in the original paper. The entire ImageNet-1K is partitioned into 50 subsets, each containing data of 20 classes. For each subset, a DiT model (Peebles & Xie, 2023) is fine-tuned for 8 epochs. The mini-batch size, representative weight and diversity weight are set as 8, 0.002 and 0.008, respectively. During inference, the corresponding fine-tuned model is loaded to generate data for specific classes.

For unCLIP Img2Img, we utilize the pre-trained model without any fine-tuning adjustments[1]. Random real images are fed into the model simultaneously with text prompts to generate high-quality

---

[1]https://huggingface.co/radames/stable-diffusion-2-1-unclip-img2img

samples without losing the information of original data distribution. We adopt 28 prompt templates for generating images, *e.g.*, "a photo of a nice {\$class_name}" (Radford et al., 2021). The utilization of text prompts for conditioning provides significant flexibility, enabling data generation for custom datasets without extra training efforts. While the absence of fine-tuning may lead to slight reduction in generation quality, it allows for direct application of the proposed CONCORD method to any custom data given relevant text descriptions. During inference, the same pre-trained model is adopted for generating images for all target categories.

**Concept Acquirement**   We use GPT-4o to retrieve descriptive concepts for different categories. The full adopted prompt is as follows:

> You are an expert in computer vision and image analysis. Here is the task: <task>I want to use some visual descriptions to identify different categories in ImageNet dataset. Please first consider whether there exist categories with similar appearance to {\$class_name}. Then please give 10 short descriptions describing the appearance features that the {\$class_name} has and can be used to distinguish it from other classes. The phrases should only focus on visual appearance of body parts or components instead of functioning. Each phrase should be detailed but also shorter than 128 characters. Each phrase starts with non-capitalized characters.</task> Give the answer in the form of <answer>["\$class_name", ["phrase1", "phrase2", "phrase3", "phrase4", "phrase5", "phrase6", "phrase7", "phrase8", "phrase9", "phase10"]]</answer>.

After retrieving the original concepts, we perform a similarity calculation between the textual concepts and real images of the corresponding category. The top 5 most similar concepts are selected for the subsequent informed diffusion process, as described in Sec. 3.2. This approach helps ensure that the selected concepts align closely with the real images, thereby enhancing the validity of the concepts used in the diffusion process to a certain extent.

**Informing**   The informing process involves similarity calculation between embeddings of images and textual concepts. We use a CLIP model with ViT-L as the visual encoder, pre-trained on LAION-2B data (Schuhmann et al., 2022) to encode these embeddings. The model weights can be downloaded from Hugging Face[2]. The generation process involves 50 denoising steps for each sample. Prior to denoising, 5 descriptive concepts from the same class as well as 10 negative concepts each from a different class are retrieved for the sample. Before extracting text embeddings, the concepts are grouped with the corresponding class name using the following format:

> {\$class_name} with {\$concept}.

During each denoising step, the similarity between the generated sample and corresponding concepts is calculated for the informing objective in Eq. 12. The informing weight $\lambda$ is set as 1 for optimal performance. The concept informing guides the denoising process to obtain completeness on essential details, and thereby enhances the instance-level quality of the generated images.

**Validation**   We adopt the validation protocol in RDED (Sun et al., 2024) to evaluate the performance of distilled data. We mainly employ a ResNet-18 (He et al., 2016) architecture for experiments, with additional ones run on ResNet-101 and ConvNets as shown in Tab. 1. Specifically, for ImageWoof, ImageNet-100, ImageNet-1K, we adopt 5-layer, 6-layer, and 4-layer ConvNets, respectively, consistent with the settings in RDED. For ImageWoof, the images are resized to $128\times128$ on ConvNet-5, while for all other cases, the images are resized to $224\times224$ for evaluation.

For ImageNet and its subsets, we employ pre-trained models[3] to generate soft labels and apply Fast Knowledge Distillation (Shen & Xing, 2022). The models are trained for 300 epochs using the AdamW optimizer, with an initial learning rate of 0.001 and a weight decay of 0.01. A cosine annealing scheduler is used to adjust the learning rate. The mini-batch size for evaluation is set the same as IPC, *e.g.*, a mini-batch size of 10 is adopted for evaluating 10-IPC sets. The applied data

---

[2]https://huggingface.co/laion/CLIP-ViT-L-14-laion2B-s32B-b82K
[3]https://github.com/LINs-lab/RDED

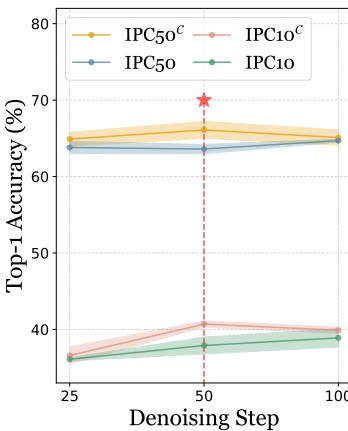

Figure 5: Parameter analysis on the diffusion denoising step number.

Table 7: Comparison with different optimization objectives and their combination on ImageWoof.

| Method | IPC | | |
|---|---|---|---|
| | 1 | 10 | 50 |
| None | $16.7_{\pm0.7}$ | $37.9_{\pm1.1}$ | $63.6_{\pm0.6}$ |
| Classifier | $16.9_{\pm0.7}$ | $38.5_{\pm1.0}$ | $65.2_{\pm0.8}$ |
| Contrastive | $\mathbf{17.4}_{\pm1.1}$ | $\mathbf{40.7}_{\pm0.4}$ | $\mathbf{66.1}_{\pm1.1}$ |
| Combination | $16.7_{\pm0.3}$ | $38.8_{\pm1.9}$ | $65.7_{\pm0.7}$ |

Table 8: Inference time cost comparison of generating one sample under the mini-batch size of 1 between the baselines and the proposed CONCORD method.

| Method | Minimax | Minimax$^{\mathcal{C}}$ | unCLIP | unCLIP$^{\mathcal{C}}$ |
|---|---|---|---|---|
| Time (s) | 2.1 | 5.3 | 9.7 | 22.8 |

augmentation techniques include patch shuffling (Sun et al., 2024), random crop resize (Wong et al., 2016), random flipping and CutMix (Yun et al., 2019). After training on the distilled dataset, the model is then evaluated with the original validation set, and Top-1 accuracy is used as the validation performance. Each experiment is performed for three times, and the mean accuracy and standard variance are reported in the results.

For the Food-101 dataset, since no pre-trained models are provided by RDED, we train a ResNet-18 model on the original training set for 300 epochs, and use it for soft-labeling. It is important to note that the utilization of pre-trained models is independent from the sample generation process, and is only for fair comparison with state-of-the-art methods, which can be omitted in actual applications.

## C    EXTENDED EXPERIMENTS AND ANALYSIS

**Ablation on Denoising Steps**    In the main experiments, we adopt 50 denoising steps for sample generation. The effect of varying the number of denoising steps is evaluated and presented in Fig. 5, with the unCLIP Img2Img model as the baseline. As the number of denoising steps increases, the accuracy of the baseline distilled data shows an upward trend under the IPC setting of 10, while is relatively consistent for the 50-IPC setting. For concept informing, fewer denoising steps result in insufficient informing, leading to performance similar to that of original images. Conversely, when too many denoising steps are used, the informing start to disrupt the standard denoising process, leading to a drop in performance. While more fine-grained tuning of the informing weight could potentially mitigate the negative effect, more denoising steps also lead to extra computational cost. Therefore, we adopt 50 denoising steps as the standard setting.

**Combining Concept Informing and Classifier Guidance**    The proposed CONCORD method informs the diffusion process to contain more discriminative details for enhancing instance-level image quality. While concept informing sharing similarity to classifier guidance, the key difference is that CONCORD utilizes the similarity between generated samples and descriptive concepts as optimization targets, without relying on pre-trained classifiers. We also conduct the experiment to combine these two kinds of constraints together to simulate scenarios where pre-trained classifiers are available. As shown in Tab. 7, when functioning independently, our proposed contrastive concept informing outperforms classifier guidance. It supports our hypothesis that detailed descriptions provide richer information compared with the category-level labels, and are more helpful in refining the instance-level sample quality. However, when both types of guidance are combined, classifier guidance does not provide additional information, and disrupts the concept informing process. As a result, the combined approach shows less effective performance improvement on the generated images. Therefore, in the main experiments, we exclusively use the proposed concept informing, as it delivers better overall results and saves extra computational consumption.

Table 9: Performance comparison with state-of-the-art methods on ImageWoof. The superscript $^{\mathcal{C}}$ indicates the application of our proposed CONCORD method. **Bold** entries indicate best results, and underlined ones illustrate improvement over baseline.

| IPC (Ratio) | Test Model | Random | K-Center | Herding | IDM | Minimax | Minimax$^{\mathcal{C}}$ | Full |
|---|---|---|---|---|---|---|---|---|
| 1 (0.08%) | ConvNet | $16.3_{\pm0.5}$ | $15.8_{\pm0.6}$ | $16.8_{\pm1.1}$ | $17.1_{\pm0.2}$ | $16.7_{\pm0.2}$ | $\mathbf{17.8}_{\pm0.8}$ | $69.0_{\pm0.2}$ |
| | ResNet-18 | $15.1_{\pm0.2}$ | $15.7_{\pm0.8}$ | $16.1_{\pm0.4}$ | $16.7_{\pm0.5}$ | $15.3_{\pm1.1}$ | $\underline{\mathbf{16.9}}_{\pm1.0}$ | $76.9_{\pm0.1}$ |
| | ResNet-101 | $14.0_{\pm0.6}$ | $13.7_{\pm1.2}$ | $14.1_{\pm0.6}$ | $16.3_{\pm0.6}$ | $14.2_{\pm1.1}$ | $\underline{\mathbf{14.9}}_{\pm1.3}$ | $77.6_{\pm0.2}$ |
| 10 (0.8%) | ConvNet | $40.5_{\pm1.5}$ | $37.1_{\pm0.9}$ | $41.2_{\pm0.4}$ | $38.5_{\pm0.6}$ | $41.2_{\pm0.8}$ | $\mathbf{43.1}_{\pm0.5}$ | $69.0_{\pm0.2}$ |
| | ResNet-18 | $34.3_{\pm1.6}$ | $33.1_{\pm0.5}$ | $36.8_{\pm0.6}$ | $36.5_{\pm1.2}$ | $42.8_{\pm1.1}$ | $\underline{\mathbf{44.4}}_{\pm0.9}$ | $76.9_{\pm0.1}$ |
| | ResNet-101 | $32.1_{\pm1.0}$ | $31.6_{\pm0.3}$ | $33.8_{\pm0.4}$ | $30.8_{\pm1.2}$ | $35.7_{\pm0.9}$ | $\underline{\mathbf{36.5}}_{\pm0.9}$ | $77.6_{\pm0.2}$ |
| 50 (3.8%) | ConvNet | $60.9_{\pm0.9}$ | $57.7_{\pm1.2}$ | $60.4_{\pm0.8}$ | $61.0_{\pm0.6}$ | $61.1_{\pm0.8}$ | $\mathbf{62.5}_{\pm0.9}$ | $69.0_{\pm0.2}$ |
| | ResNet-18 | $67.1_{\pm1.0}$ | $64.3_{\pm0.9}$ | $67.6_{\pm0.5}$ | $64.9_{\pm0.6}$ | $67.8_{\pm0.5}$ | $\underline{\mathbf{69.2}}_{\pm1.0}$ | $76.9_{\pm0.1}$ |
| | ResNet-101 | $61.4_{\pm0.7}$ | $58.8_{\pm0.4}$ | $60.8_{\pm0.3}$ | $57.2_{\pm0.6}$ | $62.2_{\pm0.6}$ | $\underline{\mathbf{63.6}}_{\pm0.2}$ | $77.6_{\pm0.2}$ |

**Extra Computational Cost** We report the inference time cost for generating an image on both Minimax and unCLIP Img2Img in Tab 8. Comparatively, introducing CONCORD increases the original inference cost by approximately 1-1.5 times. unCLIP Img2Img involves Stable Diffusion v2-1 model (Rombach et al., 2022), which demands more computational resources compared with Minimax, which uses a DiT model (Peebles & Xie, 2023) as the denoising backbone. During inference, Minimax with CONCORD only requires about half the time of unCLIP baseline. Although Minimax performs better as a baseline, the advantage is based on extra fine-tuning processes on the target dataset. Therefore, the model choice in real-world applications should consider multiple factors, including the balance between training and inference time consumption.

**Comparison to More Baselines** In addition to the results in Tab. 1, we also conduct experimental comparison with random sampling, K-Center (Sener & Savarese, 2018), Herding (Welling, 2009) and IDM (Zhao et al., 2023) in Tab. 9. For the methods based on original samples, we first resize the images to 128×128 for ConvNet and 224×224 for ResNet before running validation.

K-Center and Herding are two methods selecting coresets from the original data, with unstable performance improvement compared with random sampling. IDM is a dataset distillation method based on distribution matching, which is effective under small IPC settings. However, as the required sample number increases, the generated images often perform worse than random selected original samples. The baseline Minimax comparatively provides more stable information condensation across different IPC settings. When combined with the proposed CONCORD method, the overall dataset quality is significantly enhanced, surparssing all other methods in terms of accuracy. Especially for ConvNet and ResNet-18 architectures, training with 50 images per class achieves less than 10% performance gap from training with the entire original set. As larger models (*e.g.*, ResNet-101) require mode data and training iterations to get good performance, there still remains certain performance margin between distilled data and original full-set.

**Feature Distribution Visualization** We provide the feature distribution comparison in Fig. 6 to illustrate the effects of our proposed CONCORD method.

Firstly, the left figure shows the t-SNE features of samples generated with and without the informing of CONCORD. CONCORD works as a training-free guidance at the inference stage, without changing the main object in the images. By refining essential details in the generated samples, CONCORD enhances instance-level conceptual completeness, and improves the overall quality of the distilled datasets. However, these detail refinements have a mild effect on the feature distribution, indicating that with an already well-structured distribution, CONCORD can further improve performance without disrupting the underlying data distribution.

Secondly, the middle figure compares the generated images with the original ones used as conditioning in unCLIP Img2Img. The generated images closely align with the original data distribution, validating their effectiveness in capturing the properties of the original dataset. It demonstrates the suitability of using these generated images for training models.

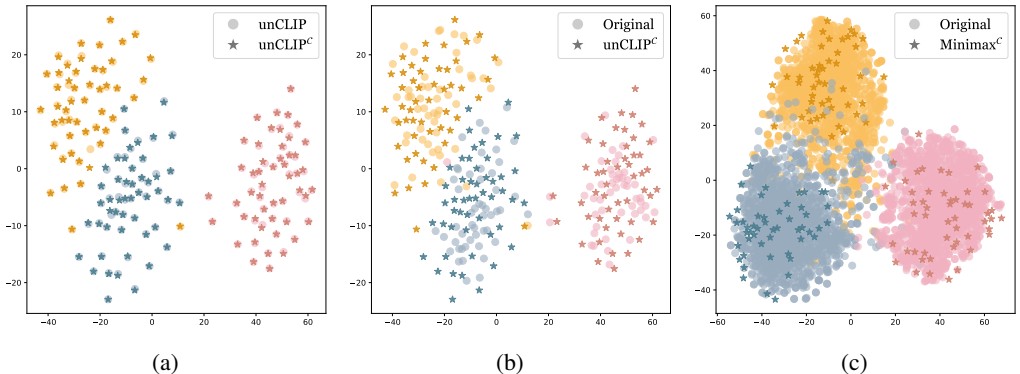

Figure 6: Feature distribution visualization of (a) samples generated by unCLIP with and without CONCORD; (b) samples generated by unCLIP with CONCORD and original samples used for conditioning; (c) samples generated by Minimax with CONCORD and original samples. Different colors indicate different categories.

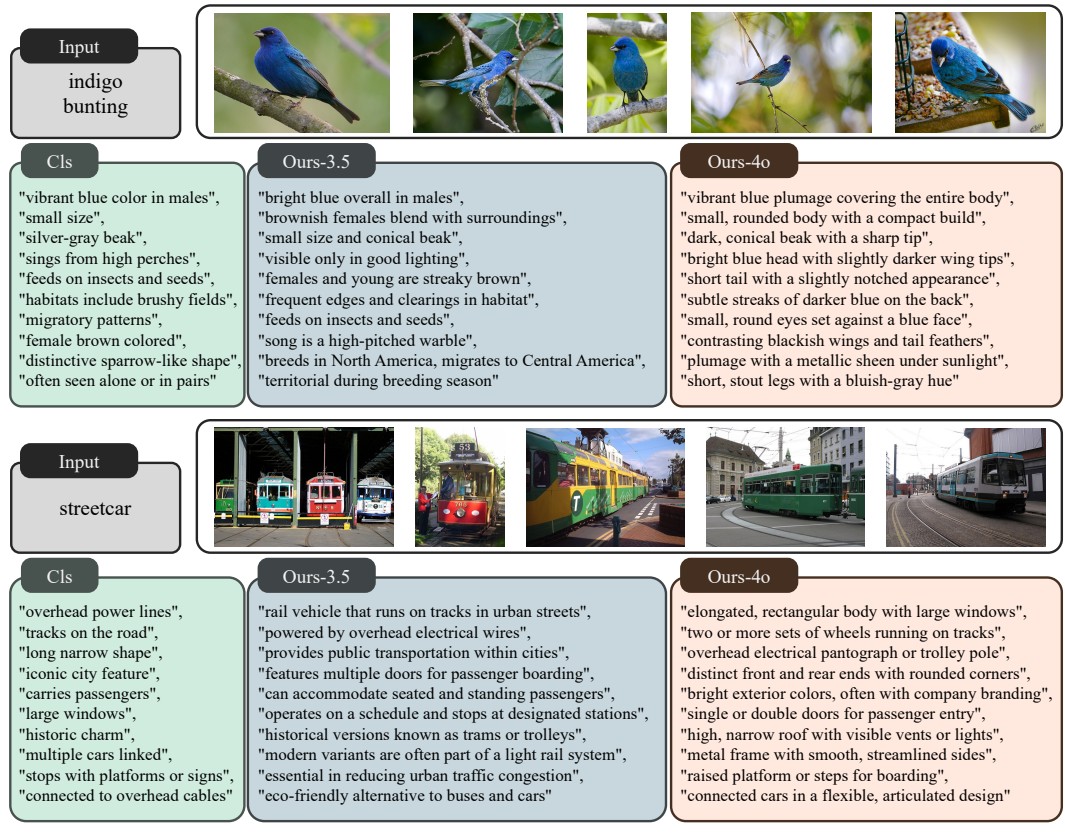

Figure 7: The comparison between concepts retrieved by different prompts and LLMs. "Cls" refers to prompts used for zero-shot classification attribute retrieval. Example images of corresponding classes are also presented to show their appearance features.

Lastly, the right figure shows the distribution of samples generated by Minimax with CONCORD and the entire original set. The generated samples demonstrate comprehensive coverage over the original distribution, ensuring that they represent a wide range of instances. While with sufficient diversity brought by samples distributed near decision boundaries, the generated samples also reduces noise in the overlapping regions between categories. It makes the generated dataset stable and effective for training models, when computational resources are limited.

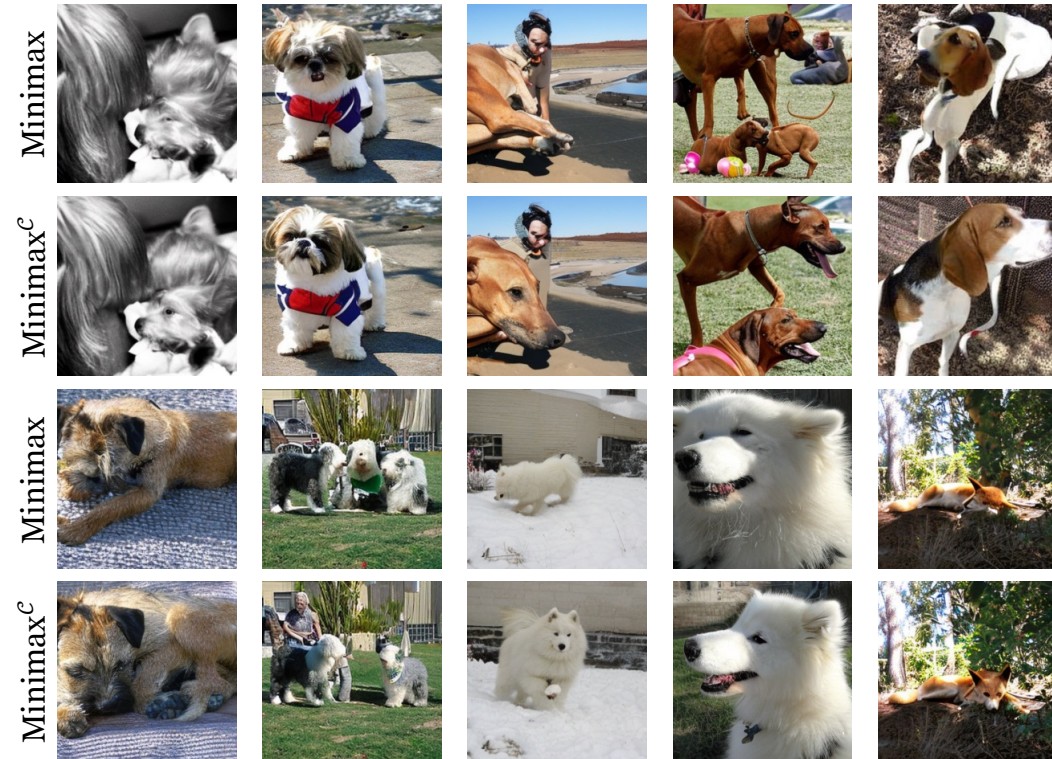

Figure 8: Example generated image comparison on Minimax with and without the proposed CONCORD method (denoted as $^{\mathcal{C}}$).

**Analysis on the Retrieved Concepts**   The effects of different prompts and LLMs have been quantitatively investigated in Tab. 4. We further conduct qualitative comparison for the retrieved descriptions to explicitly analyze the informing effects of different concepts. As shown in Fig 7, descriptions of indigo bunting and streetcar are retrieved based on three settings: prompt for zero-shot classification on GPT-4o (denoted as "Cls"), our adopted prompt on GPT-3.5, and our prompt on GPT-4o. The "Cls" prompt retrieves general descriptions about the object. However, in many cases the retrieved descriptions are still too coarse for fine-grained informing. The descriptions retrieved by GPT-3.5 are more detailed, but contain a large number of non-visual attributes, which cannot provide valid signal during concept informing. Comparatively, our adopted prompt on GPT-4 successfully emphasizes the detailed visual features of corresponding categories. These fine-grained descriptions enables the proposed CONCORD method to effectively enhance instance-level conceptual completeness and further improves the overall quality of the distilled datasets.

# D   SAMPLE COMPARISON

We further present more example generated images in the following sections.

**Comparison with Baselines**   Firstly, we show comparison on Minimax and unCLIP Img2Img with and without applying the proposed CONCORD method in Fig. 8 and Fig. 9, respectively. When baseline methods fail to present essential features and often lead to image defects, CONCORD significantly enhances the conceptual completeness in samples.

**Failure Cases**   We also present failure cases where the proposed CONCORD method fails to correct or supplement essential features in the images in Fig. 10. It can be seen that the informing tries to modify some defects in the original image, but the eventual refinement is limited. There are also some cases where the informing fails to find the missing or incorrect details. Especially the informing fails to refine the details when the number of body parts is incorrect or the body part is

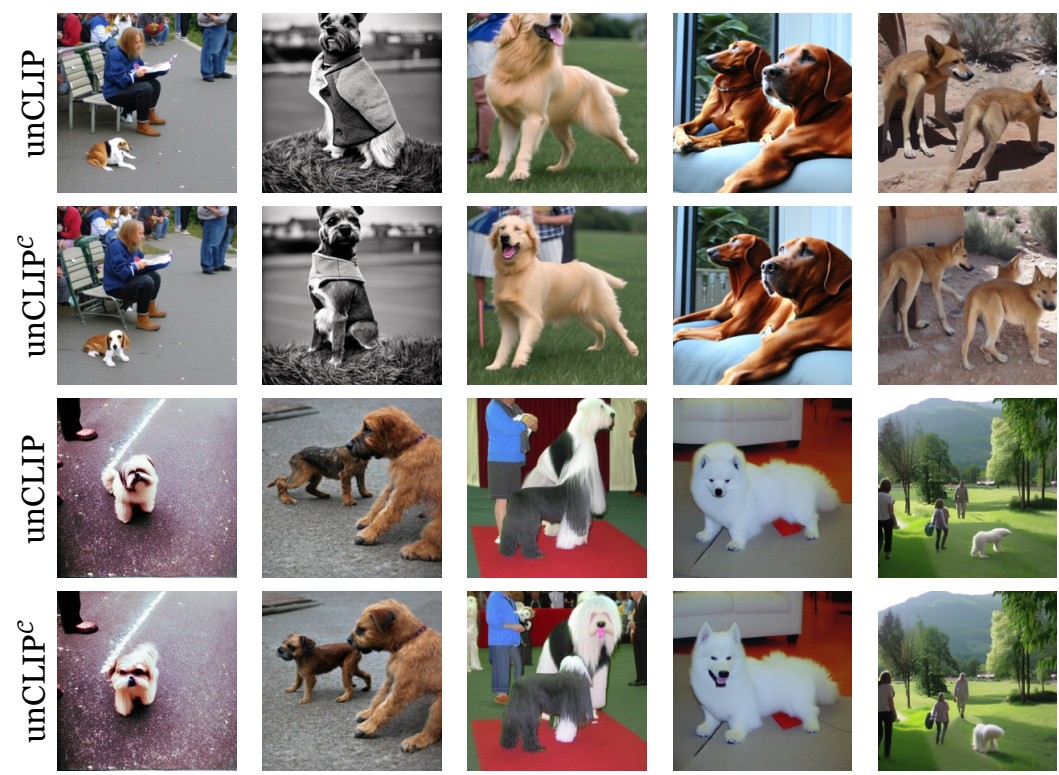

Figure 9: Example generated image comparison on unCLIP Img2Img with and without the proposed CONCORD method (denoted as $^{\mathcal{C}}$).

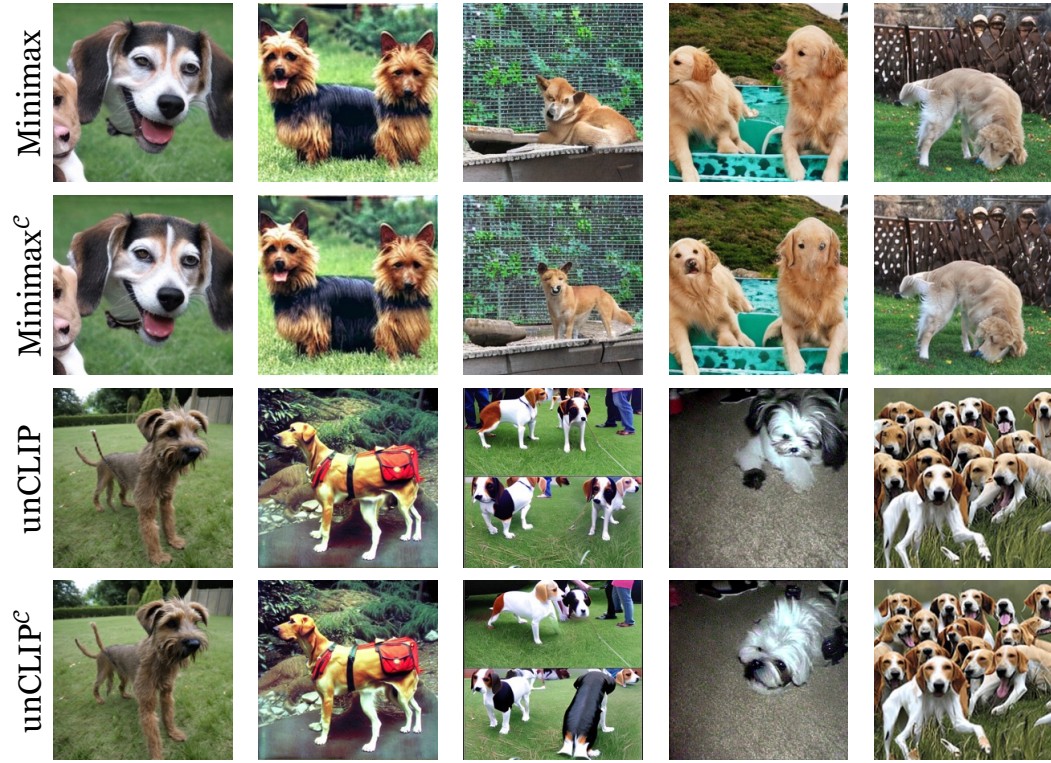

Figure 10: Example cases where CONCORD fails to supplement or modify incorrect concepts in the images ($^{\mathcal{C}}$ indicates the application of CONCORD).

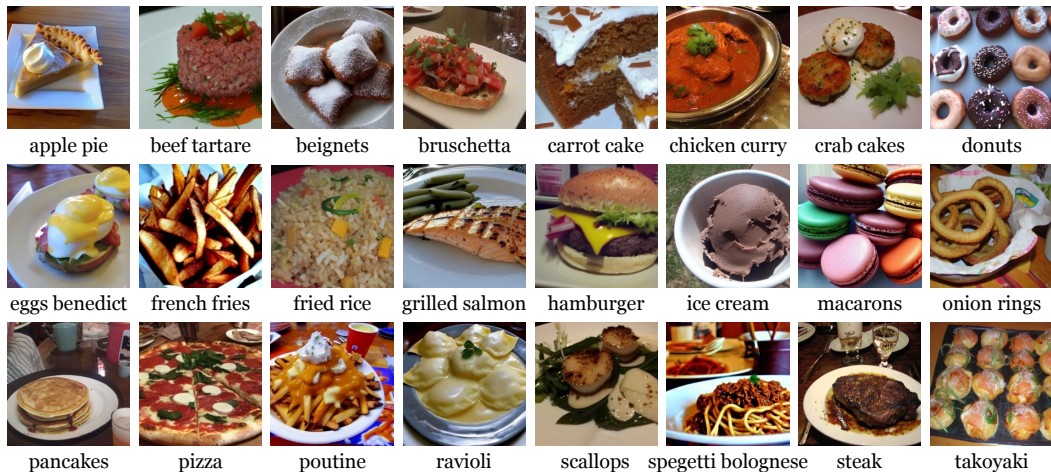

Figure 11: Example images generated by the proposed CONCORD method on the Food-101 dataset. The class names are annotated below the images.

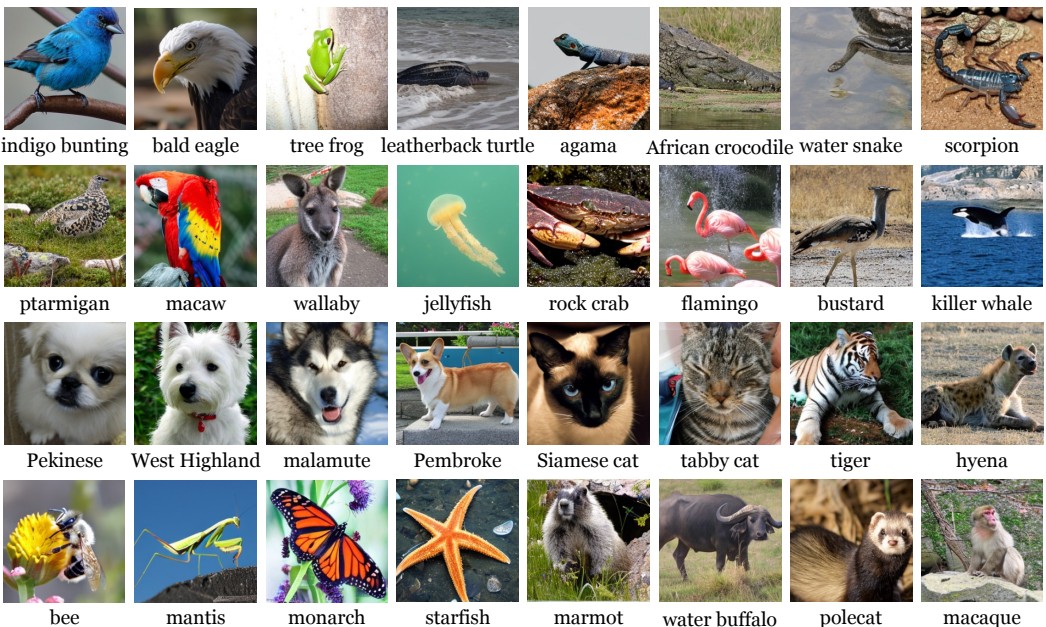

Figure 12: Example animal images generated by the proposed CONCORD method on the ImageNet-1K dataset. The employed diffusion pipeline is the fine-tuned Minimax model. The class names are annotated below the images.

completely missing in the original generation results. There is still much space for further improving the instance-level sample quality for dataset distillation.

**More Sample Visualization** Additionally, we present more example samples across various categories to demonstrate the overall high quality of the dataset generated by the proposed CONCORD method. Specifically, in Fig. 11 we present samples generated for the Food-101 dataset. In Fig. 12 images of animal categories in the ImageNet-1K dataset are generated by Minimax with CONCORD applied. In Fig. 13 we show images of other categories in the ImageNet-1K dataset. The high-quality generated samples form an effective surrogate dataset, which achieves state-of-the-art performance.

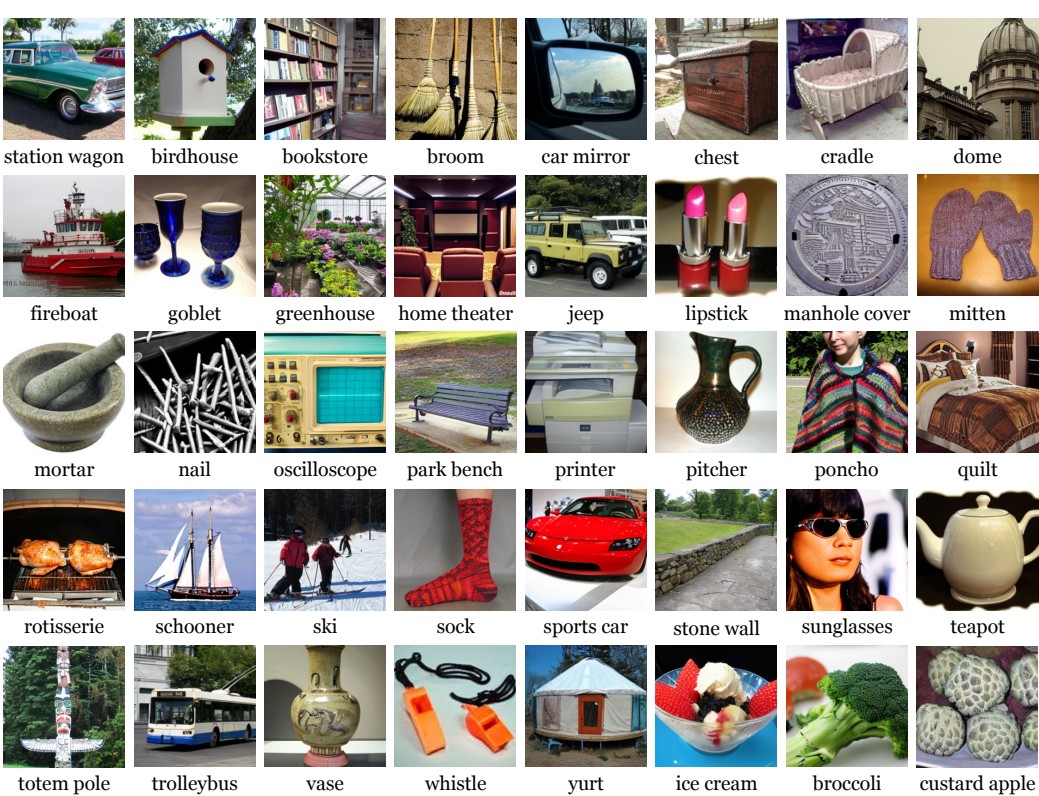

Figure 13: Example other images generated by the proposed CONCORD method on the ImageNet-1K dataset. The employed diffusion pipeline is the fine-tuned Minimax model. The class names are annotated below the images.

