# OpenReview forum: "CONCORD: Concept-informed Diffusion for Dataset Distillation"
_ICLR.cc/2025/Conference — ICLR 2025 Conference Withdrawn Submission_

### Official Review · Reviewer_7vzD · 2024-10-23

**Soundness:** 2
**Presentation:** 2
**Contribution:** 3
**Rating:** 5
**Confidence:** 4

**Summary:**

This paper introduces CONCORD, a novel dataset distillation method that leverages large language models (LLMs) to guide the diffusion process for image generation. It incorporates contrastive loss to provide adequate guidance. It improves	 both controllability and interpretability without relying on pre-trained classifiers.

**Strengths:**

The paper's visualizations clearly demonstrate improved image detail when using CONCORD.

The overall idea is innovative, as it utilizes LLM-generated content as prompts to guide the diffusion process during dataset distillation, showcasing novelty.

CONCORD allows for explicit control during the diffusion process, offering a more interpretable approach compared to traditional dataset distillation methods.

The theoretical foundations are well-established, and the inclusion of code is a pleasant surprise.

The inclusion of the code is a pleasant surprise.

**Weaknesses:**

The performance improvements are quite limited, with some of the gains potentially attributable to variance.

This method depends on the quality of concepts retrieved from LLMs; if the descriptions are not sufficiently accurate or detailed, the quality of the generated datasets could suffer.

The introduction of LLMs and contrastive loss increases the complexity of training.

**Questions:**

The paper exceeds the 10-page limit, which is a possible concern and will negatively impact the evaluation score.

For some datasets and models, the improvements are minimal. For example, in Table 1 (IPC1, ResNet-101), the variance is greater than 1, yet the improvement is less than 1.

It might be useful to explore descriptions generated by a variety of LLM models for greater diversity.

Relying solely on visualizations to demonstrate effectiveness could be strengthened by comparing loss changes with and without CONCORD.

---

> ### Author Response · Authors · 2024-11-13
> **Clarification on the page limit**
>
> Thank you for the detailed and constructive comments!
>
> We want to clarify that the reproducibility statement doesn't count toward the page limit according to the instructions (https://iclr.cc/Conferences/2025/AuthorGuide).
>
> As the reviewer mentioned "is a possible concern and will negatively impact the evaluation score", we assume this is one major factor that the reviewer gives us a score of 5. With this clarification, we would like to know whether the reviewer will reconsider the rating.
>
> Best,
> Authors

---

### Official Review · Reviewer_FF7z · 2024-10-29

**Soundness:** 2
**Presentation:** 3
**Contribution:** 2
**Rating:** 5
**Confidence:** 3

**Summary:**

This paper introduces a new technique for diffusion-based data distillation by incorporating conceptual information. Specifically, large language models are employed to identify class concepts, which are then used to enhance the diffusion process for data distillation. Experimental results demonstrate that this method consistently improves the performance of data distillation.

**Strengths:**

1. The proposed method is simple and easy to implement, making it easily integrable into existing generative-model-based data distillation approaches. Additionally, the use of large language models (LLMs) is straightforward and compatible with more advanced LLMs.

2. The experimental results show consistent improvements across multiple benchmark datasets. While the method does not outperform state-of-the-art techniques, it enhances the performance of baseline methods in nearly all cases.

**Weaknesses:**

1. The method appears heuristic, combining LLMs with diffusion models. Specifically, the concept-informed diffusion is based on the classifier-guided diffusion model, where the formulation is derived from conditional probabilities. However, this paper directly alters how conditional information is incorporated, replacing classifier guidance with concept information (the gradient of the loss function), which seems questionable. Providing explanations or justifications for these modifications in the formulas would be beneficial.

2. The claims are not fully substantiated by the experiments. In the introduction, the paper makes several claims about the advantages of the method, such as offering personalization in data distillation. This concept is unclear, and there is no experimental evidence to support it. Including additional evidence or examples would strengthen the validity of these claims.

**Questions:**

NA

---

### Official Review · Reviewer_Z8j7 · 2024-10-29

**Soundness:** 2
**Presentation:** 3
**Contribution:** 3
**Rating:** 5
**Confidence:** 4

**Summary:**

"CONCORD: Concept-informed Diffusion for Dataset Distillation" is the first to apply LLMs to dataset distillation. This approach leverages the conceptual knowledge from LLMs to guide the diffusion model, achieving a certain degree of instance-level control over image details and demonstrating significant effectiveness.

**Strengths:**

- This method demonstrates significant innovation, being the first to combine LLMs with Dataset Distillation (DD). It leverages the vast knowledge base of LLMs and guides the dataset distillation process through conceptual knowledge.

- The method greatly enhances instance-level control, addressing the issue of insufficient detail control in existing approaches to a certain extent.

- By using CLIP to verify the correlation between concepts and images, the method ensures the validity and accuracy of the concepts applied.

- The approach incorporates the idea of contrastive matching, minimizing the similarity between generated samples and negative concepts, thus improving the stability and accuracy of the generation process.

**Weaknesses:**

- The experimental results are highly dependent on the concept information provided by LLMs, which may lead to instability in performance.

- The performance improvements are limited, and there is a lack of detailed comparison with other methods in cross-architecture evaluations.

- The introduction of contrastive matching and concept evaluation may increase computational costs.

**Questions:**

- Due to the reliance on concept information, you can evaluate the performance of different LLMs or different numbers/types of concepts, or analyze the sensitivity of the results to changes in the retrieved concepts.

- We observed that using CONCORD significantly increases computation time, approximately 2-3 times that of without CONCORD. Could you explain in detail why the computation time increases so much? Are there any potential ways to effectively optimize this issue?

- Why did you not compare your method with other diffusion + DD approaches, such as D$^4$M: Dataset Distillation via Disentangled Diffusion Model, and Efficient Dataset Distillation via Minimax Diffusion?

---

> ### Author Response · Authors · 2024-11-13
> **Clarification on the experiments**
>
> Thank you for the detailed and constructive comments!
>
> We want to first clarify that the "Efficient Dataset Distillation via Minimax Diffusion" paper serves as one of the baselines in this paper, which is denoted as "Minimax" in tables. And our method shows improvement over the baseline, achieving state-of-the-art accuracy.
>
> As it is listed in the questions, we assume this is one major factor that the reviewer gives us a score of 5. With this clarification, we would like to know whether the reviewer will reconsider the rating.
>
> Best,
>
> Authors

---

### Official Review · Reviewer_qApM · 2024-11-03

**Soundness:** 3
**Presentation:** 3
**Contribution:** 1
**Rating:** 3
**Confidence:** 4

**Summary:**

This paper presents a study on enhancing the performance of knowledge distillation through a diffusion model by incorporating a concept-informing process. Using a large language model (LLM), a concept text is extracted, and during the denoising phase, CLIP-based image-text matching gradients are applied to embed concepts that accurately represent specific classes within a compressed dataset. Both positive and negative concepts are utilized to achieve a contrastive effect, refining the model’s ability to capture essential class-specific concepts.

**Strengths:**

- The proposed method utilizes diffusion models and CLIP to generate condensed images without the need for additional model training, streamlining the process and saving computational resources.
- The method successfully automates the concept extraction process through the use of large language models (LLMs), enhancing efficiency and reducing the reliance on manual intervention.
- The paper demonstrates reliable performance gains when integrated as an add-on to existing methods, showcasing its versatility and compatibility.
- It delivers consistently strong results across both fine-grained and regular datasets, highlighting its robustness and adaptability to varying data complexities.

**Weaknesses:**

- The proposed approach primarily leverages backpropagation within the CLIP feature space, which gives the impression of being a minimal extension of existing diffusion-based methods with CLIP feature matching. Rather than integrating concept-informed insights, it appears to be focused on distilling CLIP's knowledge directly.
- The method demonstrates unstable performance on IPC 1, raising concerns about its scalability to larger datasets or its generalizability across different dataset sizes with more class numbers.
- The paper lacks comparative experiments with widely recognized open-source methods, such as MTT and SRe2L, which could provide a clearer benchmark of the proposed approach’s performance.

**Questions:**

- The CLIP feature matching gradient in Equation 9 does not seem to correlate with the alpha parameter in the diffusion process, raising questions about the theoretical or empirical basis for this choice.
- The LLM experiments appear to be restricted to closed models, which may limit the generalizability and applicability of the findings. Has this method been tested with open models?
- Table 6 indicates that the proposed method may not inherently require an Img2Img structure, and even shows that DiT yields better performance. Could you clarify the rationale for selecting the current Img2Img-based structure?

---

> ### Author Response · Authors · 2024-11-13
> **Clarification on the experiments**
>
> Thank you for the detailed and constructive comments!
>
> We want to first clarify that the comparison with MTT and SRe2L is already included in Table 1. And the proposed method shows a clear advantage over these two methods in large-IPC settings.
>
> As it is listed in the weaknesses section, we assume this is one major factor that the reviewer gives us a score of 3. With this clarification, we would like to know whether the reviewer will reconsider the rating.
>
> Best,
>
> Authors

---

> ### Author Response · Authors · 2024-11-15
>
> Dear reviewer,
>
> Thanks again for your detailed reviews.
>
> In addition to the previous clarification, we would also like to reply to some other weaknesses and concerns:
> 1. For weaknesses 1, we intend to integrate the text concepts into the diffusion denoising process. CLIP here only serves as a tool to bridge the embedding space of text and images. It is true that we rely on CLIP's knowledge of the common embedding space, but I cannot see why it is a weakness. Instead of distilling CLIP's knowledge, we are distilling the corresponding text concepts into the images with the help of CLIP.
> 2. For question 3, the discussion of model choices is presented in Section B of the supplementary material. Please kindly refer to it.
>
> Best,
>
> Authors

---

### Note · Authors · 2024-11-15

I have read and agree with the venue's withdrawal policy on behalf of myself and my co-authors.